# Adverse pregnancy and birth outcomes associated with *Mycoplasma hominis, Ureaplasma urealyticum* and *Ureaplasma parvum*: a systematic review and meta-analysis

Marinjho Emely Jonduo [1,2] Lisa Michelle Vallely [1] Handan Wand [3]
Emma Louise Sweeney,[4] Dianne Egli-Gany [5] John Kaldor,[1]
Andrew John Vallely [1,2] Nicola Low [5]

For numbered affiliations see end of article.

**Correspondence to**
Professor Nicola Low;
nicola.low@unibe.ch

## ABSTRACT

**Objectives** *Mycoplasma hominis, Ureaplasma urealyticum* and *Ureaplasma parvum* (genital mycoplasmas) commonly colonise the urogenital tract in pregnant women. This systematic review aims to investigate their role in adverse pregnancy and birth outcomes, alone or in combination with bacterial vaginosis (BV).

**Methods** We searched Embase, Medline and CINAHL databases from January 1971 to February 2021. Eligible studies tested for any of the three genital mycoplasmas during pregnancy and reported on the primary outcome, preterm birth (PTB) and/or secondary outcomes low birth weight (LBW), premature rupture of membranes (PROM), spontaneous abortion (SA) and/or perinatal or neonatal death (PND).

Two reviewers independently screened titles and abstracts, read potentially eligible full texts and extracted data. Two reviewers independently assessed risks of bias using published checklists. Random effects meta-analysis was used to estimate summary ORs (with 95% CIs and prediction intervals). Multivariable and stratified analyses were synthesised descriptively.

**Results** Of 53/1194 included studies, 36 were from high-income countries. In meta-analysis of unadjusted ORs, *M. hominis* was associated with PTB (OR 1.87, 95% CI 1.49 to 2.34), PROM, LBW and PND but not SA. *U. urealyticum* was associated with PTB (OR 1.96, 95% CI 1.14 to 1.39), PROM, and SA. *U. parvum* was associated with PTB (1.79, 95% CI 1.28 to 2.52) and PROM. Seven of 53 studies reported any multivariable analysis. In two studies, analyses stratified by BV status showed that *M. hominis* and *U. parvum* were more strongly associated with PTB in the presence than in the absence of BV. The most frequent source of bias was a failure to control for confounding.

**Conclusions** The currently available literature does not allow conclusions about the role of mycoplasmas in adverse pregnancy and birth outcomes, alone or with coexisting BV. Future studies that consider genital mycoplasmas in the context of the vaginal microbiome are needed.

**PROSPERO registration number** CRD42016050962.

## STRENGTHS AND LIMITATIONS OF THIS STUDY

⇒ We followed a published protocol with predefined outcomes and statistical analysis plan.
⇒ Two reviewers independently selected the studies, extracted data and performed risk of bias assessment.
⇒ Evidence for heterogeneity was examined and described both visually and statistically.
⇒ We triangulated findings across study designs.
⇒ Restriction to studies in English and German might have missed eligible articles.

## INTRODUCTION

*Mycoplasma hominis, Ureaplasma parvum* and *Ureaplasma urealyticum,* referred to together as genital mycoplasmas, commonly colonise the urogenital tract in women, and are often found together.[1 2] These species do not appear to cause symptoms or harmful effects in non-pregnant women.[2 3] Plummer *et al* found that *M. hominis* was associated with abnormal vaginal discharge only in non-pregnant women who also had bacterial vaginosis (BV).[2] Colonisation with a genital mycoplasma has, however, been reported in many studies to be associated with several adverse pregnancy outcomes,[4 5] including preterm birth (PTB); low birth weight (LBW); premature rupture of membranes (PROM) and preterm premature rupture of the membranes (PPROM), spontaneous abortion (SA) and perinatal or neonatal death (PND).[1 6–12] Several research groups have suggested that *M. hominis,* while considered a part of the normal vaginal microbiota, might only be pathogenic in the presence of BV as part of a disturbed vaginal microbiota.[4 5 13] There are, however, inconsistencies across studies, uncertainty about

the interplay between specific organisms and the vaginal microbiota in general,[14–16] and differences in recommendations for testing and treatment.[13 17]

Technological advances in the molecular detection of multiple vaginal and endocervical organisms in the same assay[18 19] should make it easier to study the role of genital mycoplasmas in adverse pregnancy outcomes. Methods to distinguish between *U. urealyticum* and *U. parvum* were not widely available before 2000,[20 21] and unspeciated *Ureaplasma* spp. detected by culture were reported together as *U. urealyticum*.[18] Narrative reviews have not fully elucidated whether the apparent pathogenicity of genital mycoplasmas in pregnancy is associated with a particular organism, concurrent infection with multiple genital mycoplasmas and other lower genital tract organisms or confounding by other demographic, clinical and behavioural factors.[4 5 13] A systematic and quantitative assessment of these questions is, therefore timely.

### Objectives
The primary objective of this study was to investigate the associations between *M. hominis*, *U. urealyticum* and/or *U. parvum* and the risk of PTB, alone and in combination with BV. Secondary objectives were to investigate associations between each genital mycoplasma and LBW, PROM, SA and PND.

### METHODS
This systematic review followed a registered protocol,[22] which covers multiple organisms, for which findings are reported elsewhere, including *Neisseria gonorrhoeae*[23] and *M. genitalium*.[24] We report our findings using the Preferred Reporting Items for Systematic Reviews and Meta-Analyses[25] (online supplemental file, A.1) and we also used methodological guidance about systematic reviews of observational studies.[26] Patients or the public were not involved in the design, or conduct, or reporting, or dissemination plans of our research.

### Eligibility criteria, information sources and search strategy
Studies were eligible if they reported on pregnant women with and without *M. hominis*, *U. urealyticum* and/or *U. parvum* and included one or more of the outcomes: PTB, LBW, PROM (preterm or term), SA and PND. Standard definitions were used for all outcomes (PTB, delivery at <37 weeks gestation; LBW, birthweight <2.5 kg; PROM, rupture of membranes prior to onset of labour; PPROM, premature rupture at <37 weeks gestation; SA, delivery at <20 weeks gestation; stillbirth (death after >20 weeks in utero); perinatal or neonatal death (PND, stillbirths and death <28 days after birth), but we used author's definitions if necessary.[22] We excluded articles published before 2000 if they reported unspeciated *U. urealyticum* alone. If they reported on *M. hominis* and *U. urealyticum*, we included the study but did not extract results about *U. urealyticum*. We included cohort, cross-sectional and case–control studies, and randomised controlled trials.

A member of the team (MEJ) searched Medline, Embase, Cumulative Index to Nursing and Allied Health Literature (CINAHL) for literature published from January 1971 to February 2021. We searched reference lists of included studies for additional potentially eligible studies but did not search grey literature sources. The searches did not include language restrictions, but we only read the full-text of articles in English and German (languages spoken by the review team). The full search strategy is in the online supplemental file (A.2). We used Endnote (V.7, Thomson Reuters) to import, deduplicate and manage retrieved records.

### Study selection and data extraction
Two reviewers (MEJ, LMV) independently screened titles and abstracts and read the full text of potentially eligible papers. Disparities were resolved by discussion or by a third reviewer (DE-G). Where multiple reports presented data from the same study population, we identified a primary record with the most detailed information but included data from other publications. Two reviewers (MEJ, LMV) extracted data independently into an online database (Research Electronic Data Capture, REDCap, Vanderbilt University, Tennessee). Disparities were resolved by discussion or by a third reviewer (DE-G, NL or ELS).

### Data extraction
Each reviewer extracted data about the study design, study setting and sociodemographic characteristics, specimen type and timing, laboratory tests, organisms tested for, outcomes reported, raw numbers of participants with and without each outcome and organism, where available, or author-reported effect size and 95% CIs. They extracted the adjusted OR (aOR, 95% CI) and recorded variables included in multivariable models, where possible. If results were described for more than one anatomical site, we used the following order of preference: vaginal or cervical swabs, amniotic fluid, placenta, urine, blood. Where there was more than one diagnostic method, we used data from nucleic acid amplification test (NAAT), then bacterial culture, followed by ELISA. The data underlying this article are available in the article and in its online supplementary material.

### Risk of bias assessments
Two reviewers (MEJ, LMV) appraised each article independently, using checklists published by the UK National Institute for Health and Care Excellence (NICE).[27 28] A qualitative judgement about internal and external validity was summarised as: all or most checklist criteria fulfilled (++), some criteria fulfilled (+) or few or no criteria fulfilled (-). We used funnel plots and the Egger test[29] to investigate evidence for publication or small study biases across studies for outcomes reported by more than nine studies.

### Data synthesis
We used Stata V.14.0 (StataCorp, College Station, Texas) for all analyses. We used the OR, with 95% CI as the

measure of association for all study designs, since the OR and risk ratio are similar for rare outcomes, as it is the case for most of the outcomes of interest. This allowed us to analyse findings from different study designs together, where appropriate.[30] We constructed 2×2 tables to calculate the OR or used the authors' calculation when raw data were unavailable. We added 0.5 to each cell in the table if there were zero observations in one cell. For each exposure–outcome pair, we examined forest plots of univariable associations visually, displaying the OR (with 95% CI) and the $I^2$ statistic, to examine between study heterogeneity. We used a random effects model to estimate a summary OR (95% CI), which is the average effect across all included studies.[31] We stratified studies by study design in forest plots and, where the stratified estimates were compatible, we estimated the overall estimated OR with its 95% CI and a prediction interval, where there were three or more studies. The prediction interval takes into account all sources of between study variability to estimate a range of values—for the OR in a new study that is similar to the types of studies included in the meta-analysis.[31] We then examined evidence from studies that also reported on BV. We described findings from analyses that were stratified by BV status, or in studies with a multivariable analysis, we reported the aOR, controlling for BV and other measured confounding variables.[26]

## RESULTS

### Study selection

Our searches identified 1194 records and we screened 641, after exclusion of duplicates (online supplemental file, figure S1). Of 215 full-text articles, we included 53 studies. Articles excluded based on title and abstract mostly concerned neonatal respiratory outcomes, chorioamnionitis and infertility. Full-text articles were excluded for various reasons (online supplemental file, figure S1).

### Study characteristics

Of the 53 studies, we identified 42 reporting on *M. hominis* (proportion detected <1%–70%), 19 reporting on *U. urealyticum* (proportion detected 0%–90%) and 14 reporting on *U. parvum* (2%–100%) and median total sample size 241, interquartile range (IQR) 145 to 688, range 61[32] to 10397[33] (table 1, online supplemental file, table S1). There were 26 cohort studies (online supplemental file, table 2.1),[1 6 8 12 15 33–53] 22 case–control studies (online supplemental file, table S2.2)[7 9–11 32 54–70] and five cross-sectional studies (online supplemental file, table S2.3).[71–75] Most studies were from high-income settings (36/53) (online supplemental file, tables S3.1-S3.3); ethnicity was reported in 23 studies, and maternal smoking in 14 (online supplemental file, table S4.1-S4.3). Most studies (50/53) stated the timing of specimen collection, and all described the laboratory tests used (online supplemental file, table S1): 27/53 bacterial culture only; 22/53 NAAT only (table 1, online supplemental file, table S1).

Of the 53 studies, 38 reported on a single microorganism (*M. hominis*, n=34; *U. urealyticum*, n=4); nine included two genital and seven reported on all three organisms (online supplemental file, figure S2). Only one study presented findings for combinations of more than one genital mycoplasma;[47] the rest presented data separately, even if they had tested for more than one organism. Ten studies reported on the presence of BV;[33 36 43 44 47 51 53 57 58 69] we report the findings of these studies in the relevant section of the results for each genital mycoplasma. Twenty-one studies reported on other sexually transmitted infections (online supplemental file, tables S4.1-S4.3), including 2/21 reporting on syphilis, 9/21 gonorrhoea, 17/21 chlamydia, 6/21 *M. genitalium*, 7/21 trichomonas and 2/21 HIV.

Table 2 summarises the meta-analyses of each exposure–outcome pair and information about genital mycoplasmas in the presence or absence of BV (online supplemental file, table S5). In most meta-analyses, heterogeneity was low or moderate. Summary findings from different study designs were compatible, so we present summary measures across all study designs (figures 1–3, and online supplemental file, figures S3.1–S3.8).

### Risk of bias within and across studies

Based on the NICE checklists,[27 28] none of the 53 studies met all or most (++/++) checklist criteria for internal and external validity, 23 studies met some (+/+)[1 7 9 11 15 36 40 41 45–47 50 52 56 57 59 60 62 64–67 70] and 17 met few or no checklist criteria (−/−)[6 8 10 12 33 38 39 42–44 49 53 54 63 68 69 75] (online supplemental tables S6.1- S6.3). Poor reporting of study methods meant that many items could not be assessed. In all study designs, control of confounding in most studies was poorly addressed or not addressed. Regression analysis for *M. hominis* (PTB, PROM, SA), *U. urealyticum* (PTB) and *U. parvum* (PTB) did not show evidence of small study effects (online supplemental file, table S7).

### Associations between *M. hominis* and adverse pregnancy outcomes

There were 42 studies with data about *M. hominis*, reporting on 66 outcomes (online supplemental file, tables S2.1–S2.3,S3.1). Of these, 30 included data about PTB.[1 6 8 10 15 32 33 36 38 40 42–46 48 50–53 57–59 63 65–67 69–71] *M. hominis* was associated with PTB in meta-analysis of unadjusted ORs (19 576 women, summary OR 1.87, 95% CI 1.49 to 2.34; $I^2$ 29.2%; prediction interval 0.98, 3.55) (figure 1). Three studies reporting a univariable association between *M. hominis* and PTB conducted multivariable analyses (table 2, online supplemental file, table S2.1-S2.2).[1 48 59] The association was attenuated in one (aOR 1.1, 95% CI 0.5 to 2.5), after controlling for obstetric factors (previous PTB, miscarriage, multiple pregnancy and cervical incompetence).[59] In two others, authors reported no association with PTB <37 weeks, but subgroup analyses showed associations with PTB <35[1] or <33[48] weeks. In one study, no numerical results were reported (online supplemental file, table S2.1).[34] In nine studies, authors also reported on

**Table 1** Summary of characteristics of studies included in the systematic review

| Characteristic | Total | M. hominis | U. urealyticum | U. parvum |
|---|---|---|---|---|
| Number of studies, n* | 53 | 42 | 19 | 14 |
| Study design, n | | | | |
| Cohort | 26 | 22 | 10 | 7 |
| Case-control | 22 | 16 | 8 | 6 |
| Cross-sectional | 5 | 4 | 1 | 1 |
| Number of women, total (median; IQR) | 37 132 (241; 145–688) | 29 989 (250; 164–765) | 10 732 (214; 114–783) | 9890 (215; 145–972) |
| Study setting, income category, n | | | | |
| High income | 36 | 27 | 14 | 12 |
| Upper-middle income | 9 | 8 | 3 | 2 |
| Lower middle-income or low | 2 | 2 | 0 | 0 |
| Not reported | 6 | 5 | 2 | 0 |
| Outcomes reported, n* | | | | |
| Preterm birth | 40 | 30 | 16 | 13 |
| Low birth weight | 7 | 6 | 2 | 1 |
| Premature rupture of membranes | 13 | 11 | 3 | 2 |
| Spontaneous abortion | 10 | 10 | 4 | 2 |
| Perinatal death | 9 | 9 | 1 | 1 |
| Specimen type, n† | | | | |
| Endocervical swab | 24 | 20 | 7 | 6 |
| Vaginal swab | 13 | 6 | 6 | 5 |
| Urine | 1 | 1 | 0 | 0 |
| Amnotic fluid | 8 | 5 | 4 | 1 |
| Placental membrane | 7 | 6 | 2 | 2 |
| Diagnostic method* | | | | |
| NAAT | 22 | 13 | 16 | 11 |
| Culture | 28 | 28 | 0 | 0 |
| Culture and NAAT | 3 | 1 | 3 | 3 |
| Other‡ | 1 | 1 | 0 | 0 |
| Bacterial vaginosis assessed, n | 11 | 10 | 1 | 1 |
| Reported presence of STI, n | 21 | 22 | 7 | 7 |
| Reported on smoking status, n | 14 | 6 | 3 | 3 |
| Reported on multiple pregnancy, n | | | | |
| Excluded | 26 | 20 | 11 | 8 |
| Included | 9 | 6 | 4 | 3 |

*The total number of studies included is 53. The totals for each organism and outcome sum to more than 53 because one study might have reported on more than one organism and outcome.
†One study used both urine and endocervical swab.
‡ELISA (with NAAT/Culture).
ELISA, Enzyme-linked immunosorbent assay; NAAT, nucleic acid amplification test; STI, sexually transmitted infection.

BV (online supplemental file, table S5).[33 36 43 44 51 53 57 58 69] In one study, the associations between *M. hominis*, BV and PTB could be examined in detail.[33] *M. hominis*, in the absence of BV, was less strongly associated with PTB (OR 1.18, 95% CI 0.91 to 1.52) than in the presence of BV (OR 1.58, 95% CI 0.94 to 2.77).

Eleven studies included data about PROM.[6 10 40 44 45 52 59 68 70 71 75] *M. hominis* was associated with PROM in meta-analysis of unadjusted ORs (4303 women, summary OR 1.94, 95% CI 1.40 to 2.70; $I^2$ 0.0%; prediction interval 1.33 to 2.83) (online supplemental file, figure S3.1). In one study with a multivariable analysis, the association was attenuated (aOR 1.1, 95% CI 0.3 to 3.7).[59] Six studies included data about LBW.[8 34 35 49 71 73] *M. hominis* was associated with LBW in meta-analysis of unadjusted ORs (2394 newborn, summary OR 1.81, 95% CI 1.29 to 2.52; $I^2$ 0.0%; prediction interval 1.12 to 2.90) (online supplemental file, figure S3.2). In one study, *M hominis* was associated with LBW in multivariable analysis, when considered as a continuous variable (reported p=0.01).[34]

**Table 2** Summary estimates, by outcome and organism, from random effects meta-analysis of unadjusted ORs, for associations between genital mycoplasmas and adverse birth outcomes, and summary of multivariable and analyses that stratify the main association by BV status

| Adverse outcome Organism | Number of studies | Summary estimate* OR (95% CI) | $I^2$, % | Prediction interval | Any multivariable analysis† | Analyses of genital mycoplasmas and adverse birth outcomes in presence and absence of BV‡ |
|---|---|---|---|---|---|---|
| Preterm birth | | | | | | |
| M. hominis | 30 | 1.87 (1.49 to 2.34) | 29.2 | 0.98, 3.55 | 3 studies[1 48 59] | MH+,BV+/PTB OR 1.58 (95% CI 0.94 to 2.77); MH+,BV−/PTB 1.18 (0.91, 1.52)[33] |
| *U. urealyticum* | 16 | 1.96 (1.14 to 3.39) | 53.1 | 0.40, 9.73 | 3 studies[1 41 47] | None reported |
| *U. parvum* | 13 | 1.79 (1.28 to 2.52) | 59.0 | 0.66, 4.85 | 2 studies[1 47] | UP−,BV−/PTB; UP+,BV−/PTB Adjusted 1.6 (1.2 to 2.1); UP−,BV+/PTB aOR 1.6 (1.1 to 2.3); UP+,BV+/PTB aOR 2.6 (1.7 to 4.0)[47] |
| Premature rupture of membrane | | | | | | |
| *M. hominis* | 11 | 1.94 (1.43 to 2.70) | 0.0 | 1.33, 2.83 | 1 study[59] | None reported |
| *U. urealyticum* | 4 | 9.87 (1.81 to 53.72) | 49.0 | 0.02, 5757.86 | 0 studies | |
| *U. parvum* | 2 | 3.19 (1.25 to 8.15) | 0.0 | NC | 0 study | |
| Low birth weight | | | | | | None reported |
| *M. hominis* | 6 | 1.81 (1.29 to 2.52) | 0.0 | 1.12, 2.90 | 1 study[34] | |
| *U. urealyticum* | 1 | 1.08 (0.08 to 14.41) | NA | NA | 0 study | |
| *U. parvum* | 0 | NA | NA | NA | 0 study | |
| Spontaneous abortion | | | | | | None reported |
| *M. hominis* | 10 | 0.93 (0.44 to 1.94) | 50.2 | 0.12, 7.14 | 0 study | |
| *U. urealyticum* | 3 | 2.43 (1.21 to 4.86) | 0.0 | 0.03, 217.73 | 0 study | |
| *U. parvum* | 2 | 1.65 (0.67 to 4.05) | 0.0 | NC | 0 study | |
| Perinatal or neonatal death | | | | | | None reported |
| *M. hominis* | 9 | 2.70 (1.31 to 5.57) | 30.4 | 0.52, 13.94 | 0 study | |
| *U. urealyticum* | 1 | 3.52 (0.14 to 87.08) | NA | NA | 0 study | |
| *U. parvum* | 1 | 2.78 (0.11 to 68.46) | | | 0 study | |

*Meta-analysis of unadjusted ORs, using random effects model.
†Details for individual studies reported in online supplemental tables 2.1–2.3.
‡Further details of analyses based on exclusion of other infections, stratification, or multivariable analyses in online supplemental table S7.
.aOR, adjusted OR; BV, bacterial vaginosis; $I^2$, heterogeneity; MH, *Mycoplasma hominis*; NA, not applicable; NC, could not be calculated; UP, *Ureaplasma parvum*; UU, *Ureaplasma urealyticum*.

In 9 studies with data about PND,[8 32 35 40 45 51 54 72 73] meta-analysis of unadjusted ORs found an association with *M. hominis* (3696 women, summary OR 2.70, 95% CI 1.31 to 4.54; $I^2$ 30.4%; prediction interval 0.52 to 13.94) (online supplemental file, figure S3.3). In 10 studies with data about SA,[6 7 11 35 36 39 40 51 54 61] there was no association with *M. hominis* in meta-analysis of unadjusted ORs (4531 women, summary OR 0.93, 95% CI 0.44 to 1.49; $I^2$ 50.2%; prediction interval 0.12 to 7.14) (online supplemental file, figure S3.4). No results of multivariable analyses were reported for PND or SA.

### Associations between *U. urealyticum* and adverse pregnancy outcomes

Nineteen studies included data about *U. urealyticum* and 27 outcomes (online supplemental file, tables S2.1–S2.3,S3.2). There were 16 studies with data about PTB.[1 10 12 15 37 38 40 41 46 47 52 55 56 60 64 74] In meta-analysis of unadjusted ORs, *U. urealyticum* was associated with PTB (6727 women, summary OR 1.96, 95% CI 1.14 to 3.39; $I^2$ 53.1%; prediction interval 0.40 to 9.73) (figure 2). Three studies reported multivariable analyses (table 2, online supplemental file, table S2.1).[1 41 47] In one, multivariable and univariable associations were similar (aOR 1.4, 95% CI 0.8 to 2.2).[47] In one, the adjusted OR was attenuated (3.4, 95% CI 1.3, 5.5).[41] In the other, no numerical results were reported.[1]

For all other outcomes, data were only available for meta-analysis of unadjusted ORs. *U. urealyticum* was associated with: PROM in 4 studies[10 37 40 52] (1372 participants, summary OR 9.87, 95% CI 1.81 to 53.72; $I^2$ 49.0%; prediction interval 0.02 to 5757.86) (online supplemental file, figure S3.5); LBW in one study[12] (22 participants, OR 1.08, 95% CI 0.08 to 14.41; SA in three studies[7 9 40] (1204 women, summary OR 2.43, 95% CI 1.21 to 4.86; $I^2$ 0.0%; prediction interval 0.03 to 217.73)

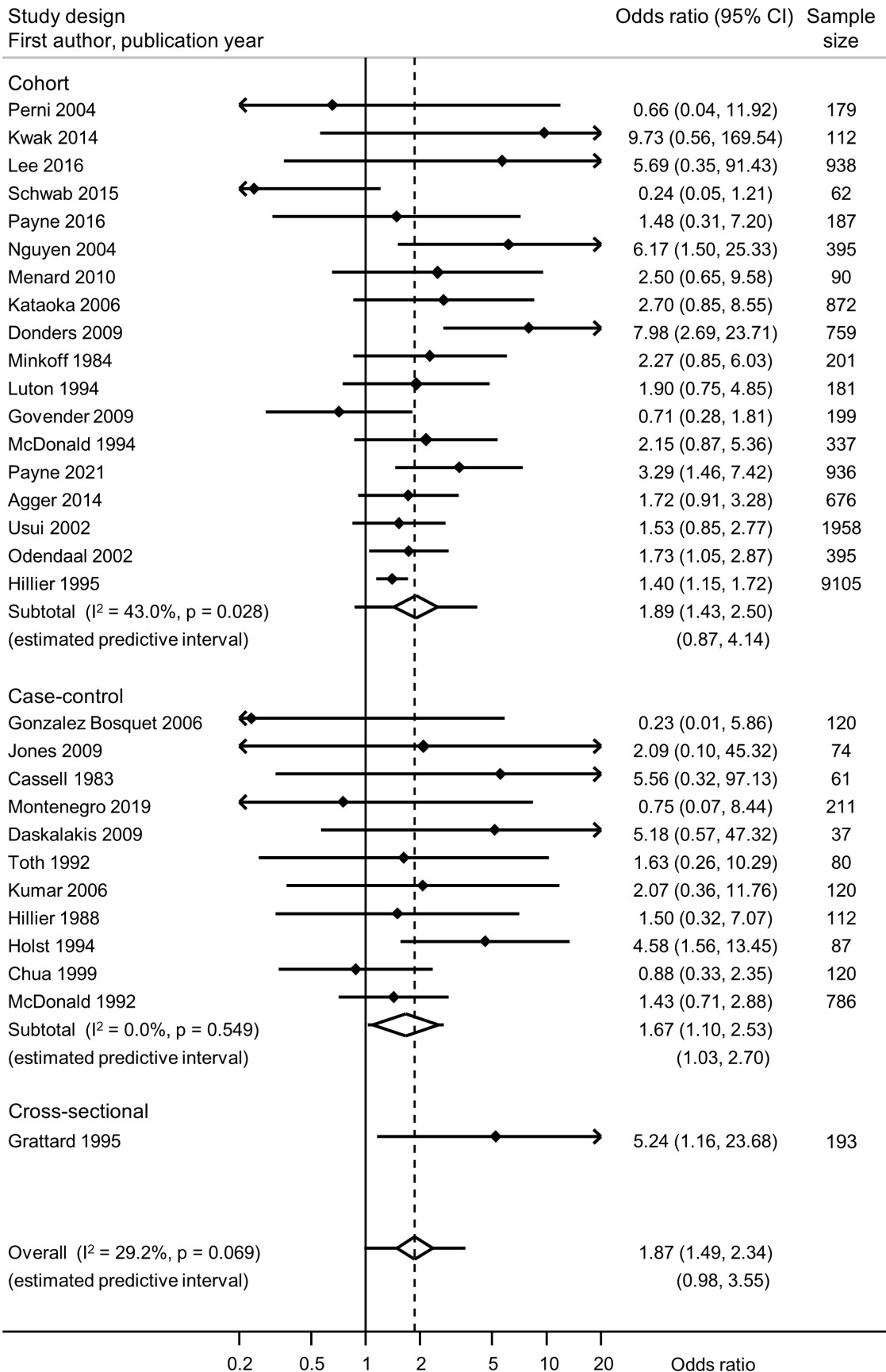

**Figure 1** Forest plot of univariable association between *M. hominis* and preterm birth, from random effects meta-analysis. Studies are in order of precision. Solid diamonds and lines either side are point estimates and 95% CIs for individual studies (arrows show where lower or upper confidence limits extend beyond the x-axis limits). Open diamond shows the point estimate and 95% CI for the summary OR and lines either side of the diamond show the prediction interval.

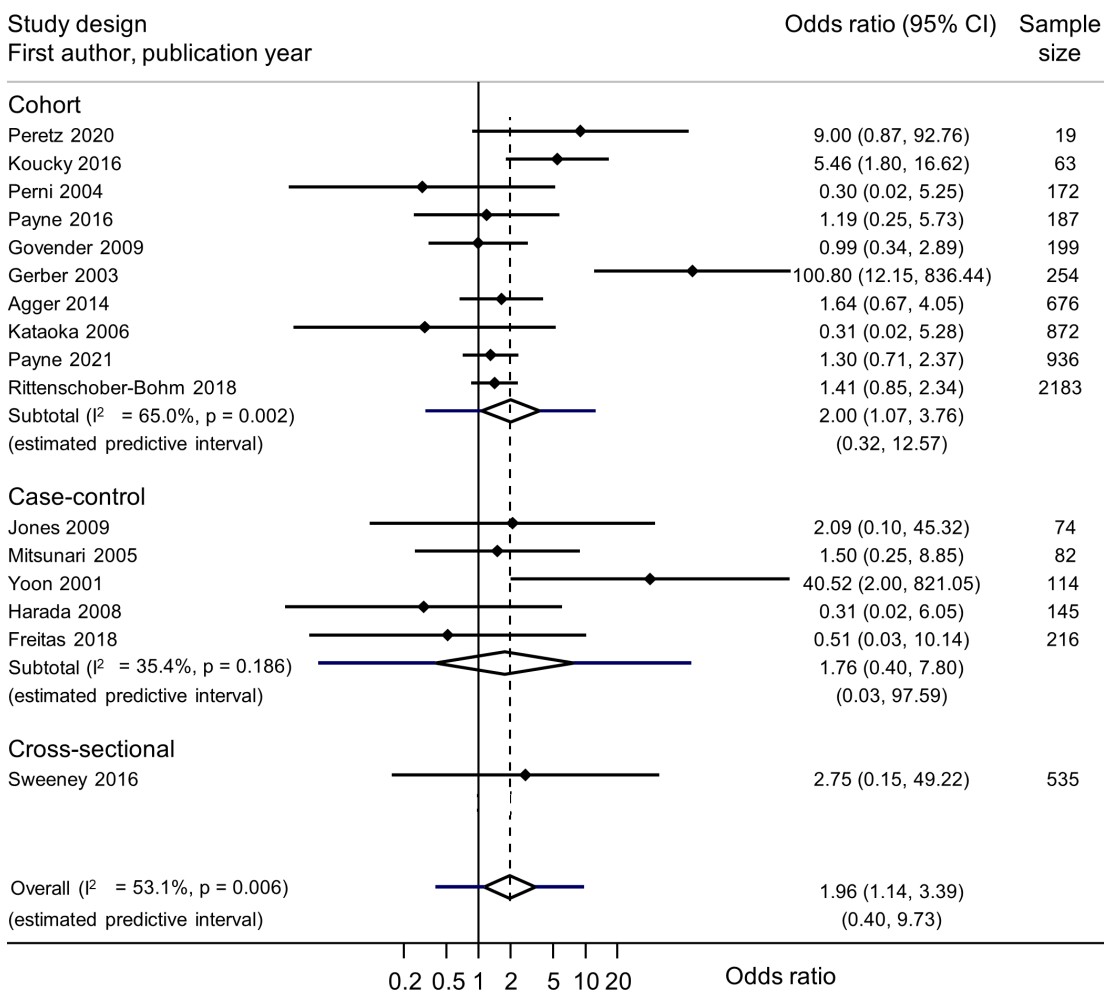

**Figure 2** Forest plot of univariable association between *U. urealyticum* and preterm birth, from random effects meta-analysis. Studies are in order of sample size. Solid diamonds and lines either side are point estimates and 95% CIs for individual studies (arrows show where lower or upper confidence limits extend beyond the x-axis limits). Open diamond shows the point estimate and 95% CI for the summary OR and lines either side of the diamond show the prediction interval.

(online supplemental file, figure S3.6) and PND in one study[40] (872 participants, summary OR 3.52, 95% CI 0.14 to 87.08).

### Associations between *U. parvum* and adverse pregnancy outcomes

Fourteen studies included data about associations between *U. parvum* and 19 outcomes (online supplemental file, tables 2.1-S2.33.1). Thirteen studies reported PTB.[1 10 12 15 38 40 46 47 55 56 60 62 74] In meta-analysis of unadjusted ORs, *U. parvum* was associated with PTB (8229 women, summary OR 1.79, 95% CI 1.28 to 2.52; I$^2$ 59.0%; prediction interval 0.66 to 4.85) (figure 3). In one study,[47] a multivariable analysis found a stronger association with PTB when both *U. parvum* and BV were present (aOR 2.6, 95% CI 1.7 to 4.0) than when *U. parvum* was present without BV (aOR 1.6, 95% CI 1.2 to 2.1), when compared with women colonised with neither (table 2, online supplemental file, table S5). In one, no numerical results were reported.[1]

For all other outcomes, data were only available for meta-analysis of unadjusted ORs. *U. parvum* was associated

with PROM in two studies[10 40] (946 participants, OR 3.19, 95% CI 1.25 to 8.15; I$^2$ 0.0%) (online supplemental file, figure S3.7) and with SA in two studies[7 40] (986 participants, summary OR 1.65, 95% CI 0.67 to 4.05; I$^2$ 0.0%) (online supplemental file, figure S3.8). One study reported on LBW (22 participants, 1 event, OR 0.56, 95% CI 0.01 to 12.75)[12] and one on PND (872 women, 1 event, OR 2.78, 95% CI 0.11 to 68.46).[40]

## DISCUSSION
### Principal findings

This systematic review and meta-analysis included 53 studies about associations between *M. hominis, U. urealyticum* and *U. parvum* and five adverse pregnancy outcomes. Only 6/53 studies reported any multivariable analysis. In 51 studies, meta-analyses of unadjusted ORs found that *M. hominis* was associated with an increase in PTB, PROM, LBW and PND, *U. urealyticum* with an increase in PTB, PROM and SA, and *U. parvum* with an increase in PTB. In two studies from which data about both genital

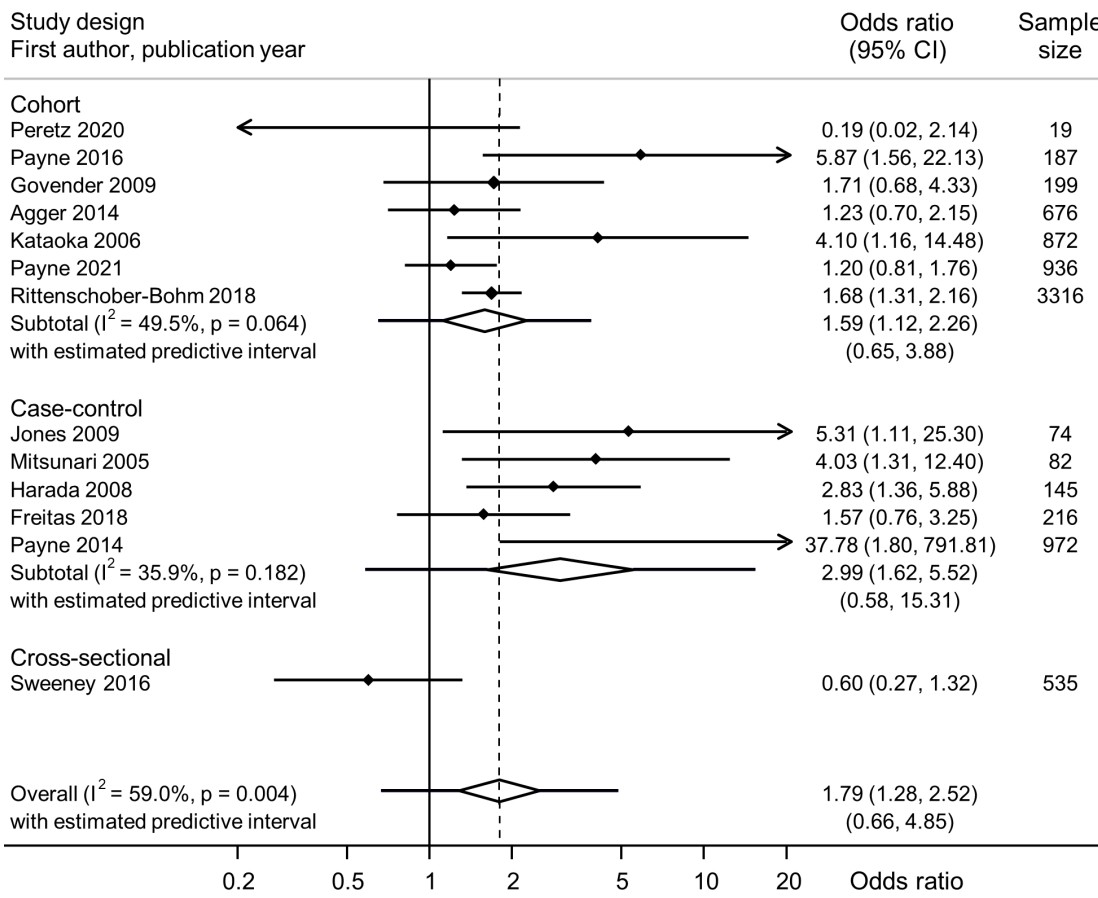

**Figure 3** Forest plot of univariable association between *U. parvum* and preterm birth, from random effects meta-analysis. Studies are in order of precision. Solid diamonds and lines either side are point estimates and 95% CIs for individual studies (arrows show where lower or upper confidence limits extend beyond the x-axis limits). Open diamond shows the point estimate and 95% CI for the summary OR and lines either side of the diamond show the prediction interval.

mycoplasmas and BV could be extracted, *M. hominis* and *U. parvum* were less strongly associated with PTB in the absence of BV than in the presence of BV.

### Strengths and weaknesses of the study

The strengths of this systematic review and meta-analysis are first, that we followed a published protocol[22] with predefined outcomes and statistical analysis plan. Study selection, data extraction and risk of bias assessment were undertaken independently by two reviewers, to reduce subjectivity. Second, we examined evidence for heterogeneity visually and statistically and calculated prediction intervals that take into account the variability in estimates from different studies and predict a range of values that could be expected in a new study.[31] In several of the random effect models, the $I^2$ value was zero, suggesting that the variability between the estimates is due to chance. This is consistent with meta-analyses in which the sampling error is high and CIs for estimates in individual studies all overlap (eg, online supplemental file, figures S3.1 and S3.2). Third, we triangulated findings across study designs;[23 26] despite the different potential sources of bias, the summary estimates were compatible and we judged it reasonable to combine effect estimates.[30] There were also limitations in the design of the review. Despite

a predefined search strategy, with broad search terms, we might have missed relevant studies, particularly by restriction to languages not spoken fluently by the authors. There were too few studies to conduct all the planned sensitivity analyses by organism, but we described all studies that allowed stratification by BV status.

### Comparison with the existing literature and interpretation

We found a systematic review about genital mycoplasmas that included studies published in English or Chinese up to March 2020.[76] The focus of the review was on infertility, however, and limited search terms for studies about adverse pregnancy outcomes identified only nine of the 53 studies that we included, making comparison difficult.

The findings from this systematic review cannot be interpreted as showing causal associations between colonisation with *M. hominis*, *U. urealyticum* or *U. parvum* in pregnancy and some adverse pregnancy outcomes. We found associations in meta-analysis of unadjusted associations, but the confounder-adjusted estimates could not be summarised. Most studies in this systematic review did not control for confounding by either sociodemographic characteristics or co-occurrence with another organism or BV. We could not elucidate the role of co-occurrence with BV,[4 5] because there were only two relevant studies, with

imprecise estimates. Rittenschober-Böhm *et al*, studied more than 4000 women in Austria.[47] They found univariable associations between both *U. parvum* (OR 1.7, 95% CI 1.3 to 2.2) and *U. urealyticum* (1.4, 95% CI 0.9 to 2.3) and spontaneous PTB. A strength of their study is the multivariable analysis, controlling for age, smoking, history of PTB and other infections. For *U. parvum,* the association with PTB was stronger when both BV and *U. parvum* were present than for *U. parvum* alone. The authors did not analyse the association with *U. urealyticum* further. Hillier *et al* investigated the association between *M. hominis* and PTB of LBW infants in more than 10 000 women in the USA.[33] The association was stronger in the presence (1.58, 95% CI 0.94 to 2.77) than absence (1.18, 95% CI 0.91 to 1.52) of BV, but CIs for both estimates include the null value. Hillier *et al* also reported a stronger association with PTB when *M. hominis* was present with Bacteroides and BV (OR 2.1, 95% CI 1.5, 3.0). The authors did not, however, control for any other confounding factors.

Several of the limitations that we found in our review apply to systematic reviews of observational studies in general. Most included studies did not set out to study our review question and have small sample sizes. We extracted most data about genital mycoplasmas, our exposures of interest, from tables of covariates. Differences in the performance characteristics of diagnostic methods might have resulted in misclassification of colonisation status. Bacteriological culture has been considered the gold standard for the identification of genital mycoplasmas, but problems can arise from their fastidious growth requirements and a lack of reliable media. Commercialised kits for both culture and NAAT diagnosis are less laborious and have greater sensitivity and specificity compared with earlier in-house approaches.[77 78] There were, however, confusions in nomenclature (eg, incorrect identification of biovar 1 as *U. urealyticum* rather than *U. parvum*)[56] [60] and misclassification of cultured ureaplasma as *U. urealyticum.*[6 50 53 65 68 70] Sample integrity is also important and greatly influenced by sample collection methods (eg, type of swab, transport medium), transportation (eg, cold chain maintenance) and storage (eg, duration and temperature at which kept in long-term storage). It was not possible to account for differences in anatomical sampling site that may have affected detection in individual studies, for example, *M. hominis* commonly colonises the lower genital tract while *Ureaplasma* spp. may colonise the upper genital tract.[79] Other limitations include misclassification, for example, gestational age was assessed by obstetric ultrasound in only one third of studies and inconsistency in the timing during pregnancy of sampling for genital mycoplasmas.

The specificity of associations between different genital mycoplasmas and adverse pregnancy and their mechanisms of action remain unclear. Several studies included in this review postulate that subclinical *Ureaplasma* spp, ascending to the choriodecidual space cross the fetal membrane is followed by placental transfer into the amniotic cavity,[7 72 74 80 81] leading to PROM, SA and PND in women with high bacterial load in the upper genital tract.[81 82] The presence of genital mycoplasmas in the placental membranes and amniotic fluid might have a direct effect, but they also increase levels of a variety of cytokines and other inflammatory mediators, which might be the key drivers of adverse pregnancy outcomes.[37 52 62 64 65 81 83] Gene sequencing methods show the diversity of the vaginal microbiota during pregnancy[15 16 84] and genital mycoplasmas are often among the most plentiful of the many bacterial species identified. In our review, one study using 16s rRNA sequencing found a group of bacteria, including *U. parvum*, that was associated with PTB,[15] but another smaller study did not.[55] Analysis of associations between microbial communities and PTB was beyond the scope of our systematic review. A better understanding of antimicrobial susceptibility is also needed. Genital mycoplasmas lack a rigid cell wall, which allows them to evade some antibiotics. Beta-lactam antibiotics and vancomycin are considered ineffective but macrolides, fluoroquinolones and tetracyclines are often effective.[85] In pregnant women requiring antimicrobials, only macrolides should be used[86] but high rates of antibiotic resistance are reported in many settings,[4 87 88] and in the absence of definitive evidence of the benefits of treatment, cannot currently be recommended.

## Implications

The findings of this systematic review show key areas for future research. First, there is a need for epidemiological studies that are designed specifically to investigate the pathogenicity of vaginal and cervical organisms alone and in the context of the vaginal microbiome. A holistic approach that includes gene sequencing and other molecular and culture methods to detect other endogenous and sexually transmitted organisms is required,[14–16] taking into account the need for consistent strategies for specimen collection both in terms of the trimester(s) and the timing and types of specimens collected. These studies should also define potential causal pathways and address confounding from factors such as maternal age, smoking, obstetric history, co-occurrence and comorbidities. Second, there is a critical need to conduct research in low-income and middle-income settings where the prevalence of sexually transmitted infections, BV and genital mycoplasmas is high, and the burden of adverse pregnancy outcomes was greatest. If consistent and reproducible associations are found in observational studies, potential interventions need to be evaluated. Randomised controlled trials of screening and treatment for a range of vaginal and endocervical infections in pregnancy are underway.[89 90] If these interventions prevent adverse pregnancy outcomes, further research will still be needed to understand the contributions of specific organisms or combinations thereof. Multiplex assays will facilitate these research studies but should not be used in routine clinical practice because of the risks of overdiagnosis and overtreatment.[18 19]

## CONCLUSIONS

In this systematic review and meta-analysis, we found that genital mycoplasmas are associated with several different

adverse pregnancy outcomes in univariable analysis only. The currently available literature does not allow conclusions about the role of genital mycoplasmas in adverse pregnancy and birth outcomes, alone or with coexisting BV. Future studies that consider genital mycoplasmas in the context of the vaginal microbiome are needed.

**Author affiliations**
¹Global Health Program, The Kirby Institute, University of New South Wales, Sydney, New South Wales, Australia
²Sexual and Reproductive Health Unit, Papua New Guinea Institute of Medical Research, Goroka, Eastern Highlands Province, Papua New Guinea
³Biostatistics and Databases Program, The Kirby Institute, University of New South Wales, Sydney, New South Wales, Australia
⁴Infectious Diseases Unit, The University of Queensland Centre for Clinical Research, Herston, Queensland, Australia
⁵Institute of Social and Preventive Medicine, University of Bern, Bern, Bern, Switzerland

**Contributors** DE-G, NL, AJV, LMV conceived the idea for the review and DE-G, JK, NL, AJV, LMV, HW wrote the protocol. MEJ and LMV did the searches, screened, and selected studies and extracted data. DE-G, NL, ELS resolved disagreements. NL and HW did statistical analyses. MEJ wrote the first draft of the manuscript. MEJ, ELS, LMV and NL did the revision and editing. All authors commented on revisions of the manuscript and accept responsibility for its content. NL is guarantor for the overall content.

**Funding** NL receives funding from the Swiss National Science Foundation, project numbers 197831, 160909; LMV is supported by an Australian National Health & Medical Research Council (NHMRC) Early Career Fellowship Grant (2018-2021); MEJ is a PhD research student supported through the Australian Award/UNSW UIPA scholarship and the Women And Newborns Trial of Antenatal Interventions and Management (WANTAIM) trial (ISRCTN No: ISRCTN37134032), funded by DFID/MRC/Wellcome Trust Joint Global Health Trials, Australian NHMRC Grant and Swiss National Science Foundation. DEG received salary support from r4d programme (Swiss Programme for Research on Global Issues for Development), grant number IZ07Z0-160909. AV receives salary support from the Australian NHMRC, through a Career Development Fellowship.

**Competing interests** NL is on the advisory board of Sefunda AG, a start-up company that develops point-of-care tests for sexually transmitted infections.

**Patient and public involvement** Patients and/or the public were not involved in the design, or conduct, or reporting, or dissemination plans of this research.

**Patient consent for publication** Not applicable.

**Ethics approval** Not applicable.

**Provenance and peer review** Not commissioned; externally peer reviewed.

**Data availability statement** No data are available. Data relevant to the study are included in the article or the supplementary information.

**ORCID iDs**
Marinjho Emely Jonduo http://orcid.org/0000-0002-8625-8488
Lisa Michelle Vallely http://orcid.org/0000-0002-8247-7683
Handan Wand http://orcid.org/0000-0002-8279-7652
Dianne Egli-Gany http://orcid.org/0000-0002-4725-0475
Andrew John Vallely http://orcid.org/0000-0003-1558-4822
Nicola Low http://orcid.org/0000-0003-4817-8986

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
