## [Reviewer comments · BMJ Open]

ARTICLE DETAILS

TITLE (PROVISIONAL)	Adverse pregnancy and birth outcomes associated with Mycoplasma hominis, Ureaplasma urealyticum and Ureaplasma parvum: A systematic review and meta-analysis
AUTHORS	Jonduo, Marinjho; Vallely, Lisa; Wand, Handan; Sweeney, Emma; Egli-Gany, Dianne; Kaldor, John; Vallely, Andrew; Low, Nicola

VERSION 1 – REVIEW

REVIEWER	Oana Maria Ionescu Carol Davila University of Medicine and Pharmacy, Obstetrics-Gynaecology
REVIEW RETURNED	03-Apr-2022

GENERAL COMMENTS	two minor observations: in abstract you used BV without explaining the meaning of this acronym (we found the explanation in introduction) in results = line 170 - "of the 52 studies" while in line 166 and in line 43 (in abstract) you specified that 57 studies are included in you analysis. please rectify this typing error
--

REVIEWER	Ana Royuela Instituto de Investigación Sanitaria Puerta de Hierro Majadahonda, Biostatistic Unit
REVIEW RETURNED	29-Apr-2022

GENERAL COMMENTS	The quality of methodology applies the standards for this type of studies. The authors have registered previously the protocol in PROSPERo, have reported the manuscript attending to the PRISMA requirements and the MOOSE guide. Analysis section is very well described and the methods applied are the proper ones for this type of research. Very well written. Some questions/minor issues: - Who performed the literature search? If a documentalist did it, it should be added into the "Eligibility criteria, information sources and search strategy" section. If was a member of the team, it also should be cited. - The authors used the prediction interval as well as the popular 95%CI. Since the prediction intervals are not as widely known as the confidence intervals, the authors should include a definition for the later in the analysis section.
--

	-Table2. How the authors could explain that having 11 studies for PROM, the I2 was equal to 0? The same is observed for LBW for M.hominis with 6 studies and SA for U. urealyticum (4 studies). - There is no coincidence between the upper limit 95%CI for U. parvum and PTB between table 2 and the corresponding text (2.27 vs 2.72). Please, verify which is correct and correct the typo.
--	--

REVIEWER	Yuki Kodama University of Miyazaki
REVIEW RETURNED	07-May-2022

GENERAL COMMENTS	Thank you for the opportunity to review this article. The authors conducted systematic review evaluating genital mycoplasmas in relation to adverse pregnancy and birth outcomes, in combination with or without bacterial vaginosis among 57 studies. The authors mentioned that the currently available literature did not allow conclusions about the role of genital mycoplasmas in adverse pregnancy and birth outcomes, alone or with co-existing bacterial vaginosis. The results obtained will be useful in the management of pregnant women having genital mycoplasmas. And I agree to the authors' comment, "... in the absence of definitive evidence of the benefits of treatment, cannot currently be recommended". It is appreciated several points below are reconsidered.  1. On page 14-16, M. hominis and U. urealyticum were associated with PTB in meta-analysis of unadjusted ORs, although both in the absence or presence of BV, they were not associated with PTB. Could you comment on this disparity? 2. In Results, on page 16, U. urealyticum was also associated with LBW (OR 2.24, 95% CI 1.16, 4.33). Please comment why the authors did not mention about it in principal findings (page 17) and in the abstract. Minor comments:  1. In page 14, line 221-222: "(Tables S3.1-S3.4)". 2. In page 15, line 254-256, ... U. urealyticum was associated with PTB (12,234 women, summary OR 1.84, 95% CI 1.34, 2.55; I2 69.2%; predictive interval 0.54, 6.36). Please reassure some figures were different from those in Figure 2. 3. In page 16, line 273-274, ... U. parvum was associated with PTB (8,002 women, summary OR 1.60, 95% CI 1.12, 2.30; I2 58.4%; predictive interval 0.59, 4.36). Please reassure some figures were different from those in Figure 3. And Figure 3 and Figure S5.1 seem similar on the association between U. parvum and preterm birth, but figures of overall were a little bit different. Please check on that.
--

VERSION 1 – AUTHOR RESPONSE

Reviewer: 1

two minor observations:

1. in abstract you used BV without explaining the meaning of this acronym (we found the explanation in introduction)

Authors response: Please see our response to the first comment from the editor.

2. in results = line 170 - "of the 52 studies" while in line 166 and in line 43 (in abstract) you specified that 57 studies are included in you analysis. please rectify this typing error

Authors response: Thank you for pointing out this typographical error. The total number is '57 studies' (line 168).

>

Reviewer: 2

> Some questions/minor issues:

1. Who performed the literature search? If a documentalist did it, it should be added into the "Eligibility criteria, information sources and search strategy" section. If was a member of the team, it also should be cited.

Authors response: We agree. In the methods, we have added, 'A member of the team (MJ) searched...' in the 2nd paragraph under search strategy section (line 114).

2. The authors used the prediction interval as well as the popular 95%CI. Since the prediction intervals are not as widely known as the confidence intervals, the authors should include a definition for the later in the analysis section.

Authors response: Thank you for the opportunity to provide additional information. We have amended the text to clarify the use of the prediction interval in paragraph 1 of the Data synthesis section_ lines 159-161 "...we estimated the overall estimated OR with its 95% CI and a prediction interval, where there were three or more studies. The prediction interval takes into account all sources of between study variability to predict a range of values- for the OR in a new study that is similar to the types of study included in the meta-analysis.[31]" We also give this explanation in the Discussion (lines 304-306).

3. Table2. How the authors could explain that having 11 studies for PROM, the I2 was equal to 0? The same is observed for LBW for M. hominis with 6 studies and SA for U. urealyticum (4 studies).

Authors response: The I² value of 0.0% is plausible and there are different reasons for this possibility. We have added text in the discussion, under 'Strengths and weaknesses,' 'In several of the random effects models, the I² value was zero, suggesting that the variability between the estimates is due to chance. This is consistent with meta-analyses in which the sampling error is high and confidence intervals for estimates in individual studies all overlap.' (lines 306–309).

4. There is no coincidence between the upper limit 95%CI for U. parvum and PTB between table 2 and the corresponding text (2.27 vs 2.72). Please, verify which is correct and correct the typo.

Authors response: Apologies for the typographical error. We have corrected the numbers in Table 2 and the text in line 276, 'OR 1.60, 95% CI 1.12, 2.30; I2 58.4%; prediction interval 0.59, 4.36) (Figure 3)'.

Reviewer: 3

It is appreciated several points below are reconsidered.

1. On page 14-16, M. hominis and U. urealyticum were associated with PTB in meta-analysis of unadjusted ORs, although both in the absence or presence of BV, they were not associated with PTB. Could you comment on this disparity?

Authors response: Thank you for the comment, which we discuss on p19-20. The apparent inconsistency probably results from the very small number of studies that stratified results by BV status. We have amended the text to clarify (lines 324-329), 'We found associations in meta-analysis of unadjusted associations, but the confounder adjusted estimates could not be summarised. Most studies in this systematic review did not control for confounding by either sociodemographic

characteristics, or co-infection with another organism or BV. We could not elucidate the role of co-infection with BV,[4, 5] because there were only two relevant studies, with imprecise estimates.’

2. In Results, on page 16, *U. urealyticum* was also associated with LBW (OR 2.24, 95% CI 1.16, 4.33). Please comment why the authors did not mention about it in principal findings (page 17) and in the abstract.

Authors response: This was an oversight, and the text ‘LBW’ has been added in Abstract (line 46) and Principal findings (line 294) accordingly.

Minor comments:

1. In page 14, line 221-222: “(Tables S3.1-S3.4)”.

Authors response: Thank you for pointing out this typographical error. The text is corrected to ‘(Tables S3.1- S3.4)’_Line 224

2. In page 15, line 254-256, ... *U. urealyticum* was associated with PTB (12,234 women, summary OR 1.84, 95% CI 1.34, 2.55; I2 69.2%; predictive interval 0.54, 6.36). Please reassure some figures were different from those in Figure 2.

Authors response: Apologies we have corrected the text, ‘1.84, 95% CI 1.34, 2.55; I2 69.2%; prediction interval 0.54, 6.36)’_Line 257-258

3. In page 16, line 273-274, ... *U. parvum* was associated with PTB (8,002 women, summary OR 1.60, 95% CI 1.12, 2.30; I2 58.4%; predictive interval 0.59, 4.36). Please reassure some figures were different from those in Figure 3. And Figure 3 and Figure S5.1 seem similar on the association between *U. parvum* and preterm birth, but figures of overall were a little bit different. Please check on that.

Authors response: We have corrected the text, ‘OR 1.60, 95% CI 1.12, 2.30; I2 58.4%; prediction interval 0.59, 4.36) (Figure 3)’ (lines 275-276) and figure S5.1 has been removed from the supporting document (updated figure numbers on page 40 and 41 of supplementary material).

VERSION 2 – REVIEW

REVIEWER	Ana Royuela Instituto de Investigación Sanitaria Puerta de Hierro Majadahonda, Biostatistic Unit
REVIEW RETURNED	27-May-2022
GENERAL COMMENTS	The authors have resolved all the minor concerns about the paper and now I recommend for this publication.
REVIEWER	Yuki Kodama University of Miyazaki
REVIEW RETURNED	10-Jun-2022
GENERAL COMMENTS	Thank you for the opportunity to review this article again. The authors conducted systematic review evaluating genital mycoplasmas in relation to adverse pregnancy and birth outcomes, in combination with or without bacterial vaginosis among 57 studies. The points which I raised were revised properly.